# Unravelling the Intricate Roles of FAM111A and FAM111B: From Protease-Mediated Cellular Processes to Disease Implications

**DOI:** 10.3390/ijms25052845

**Published:** 2024-02-29

**Authors:** Danielle Naicker, Cenza Rhoda, Falone Sunda, Afolake Arowolo

**Affiliations:** 1Division of Medical Biochemistry, Department of Integrative Biomedical Sciences, University of Cape Town, Cape Town 7925, South Africa; nckdan004@myuct.ac.za; 2Hair and Skin Research Unit, Division of Dermatology, Department of Medicine, University of Cape Town, Cape Town 7925, South Africa; cenza.rhoda@uct.ac.za (C.R.); sndfal001@myuct.ac.za (F.S.); 3Biomedical Research and Innovation Platform, South African Medical Research Council, Cape Town 7500, South Africa

**Keywords:** FAM111A, FAM111B, proteases, genetic fibrosing disorders

## Abstract

Proteases are critical enzymes in cellular processes which regulate intricate events like cellular proliferation, differentiation and apoptosis. This review highlights the multifaceted roles of the serine proteases FAM111A and FAM111B, exploring their impact on cellular functions and diseases. FAM111A is implicated in DNA replication and replication fork protection, thereby maintaining genome integrity. Additionally, FAM111A functions as an antiviral factor against DNA and RNA viruses. Apart from being involved in DNA repair, FAM111B, a paralog of FAM111A, participates in cell cycle regulation and apoptosis. It influences the apoptotic pathway by upregulating anti-apoptotic proteins and modulating cell cycle-related proteins. Furthermore, FAM111B’s association with nucleoporins suggests its involvement in nucleo-cytoplasmic trafficking and plays a role in maintaining normal telomere length. FAM111A and FAM111B also exhibit some interconnectedness and functional similarity despite their distinct roles in cellular processes and associated diseases resulting from their dysfunction. FAM111A and FAM111B dysregulation are linked to genetic disorders: Kenny–Caffey Syndrome type 2 and Gracile Bone Dysplasia for FAM111A and POIKTMP, respectively, and cancers. Therefore, the dysregulation of these proteases in diseases emphasizes their potential as diagnostic markers and therapeutic targets. Future research is essential to unravel the intricate mechanisms governing FAM111A and FAM111B and explore their therapeutic implications comprehensively.

## 1. Introduction

Proteases assume a pivotal enzymatic role, encompassing a broad spectrum of intricate cellular processes such as cellular proliferation, differentiation and the orchestration of programmed cell death [1,2]. They recognize specific protein or peptide substrates and cleave amide bonds in a highly regulated manner. Protease activity relies on a multifaceted interplay of factors, including small molecules, cofactors, post-translational modifications and, occasionally, initial cleavage of an inactive zymogen by another protease [2]. This elaborate regulatory framework serves as a robust mechanism to ensure tight control over enzyme function. Proteases are categorized into six classes based on their proteolytic mechanisms: serine, threonine, cysteine, glutamic, aspartic and metalloproteases [3]. Among these, serine proteases, which derive their name from the pivotal catalytic serine residue, constitute around a third of the protease population in homo sapiens [3]. FAM111A, also known as FAM111 trypsin-like peptidase protein, is a serine protease recently explored for its role within human cells [4]. It is a 611 amino acids protein with a molecular weight of about 70 kDa. The C-terminal region of the human FAM111A protein harbors a serine protease domain reminiscent of the Trypsin 2 enzyme, complete with a conserved catalytic triad composed of His385, Asp439 and Ser541 (Figure 1). Recently, researchers confirmed FAM111A’s protease activity by conducting in vitro experiments with its recombinant form [5]. Its auto-catalytic event was elaborated upon in cells overexpressing FAM111A, pinpointing a significant cleavage site between Phe334 and Gly335 [5]. FAM111A encompasses additional structural domains, which include a PCNA (proliferating cell nuclear antigen) interacting peptide (PIP) box [6] and two ubiquitin-like domains (UBL-1 and UBL-2) [7]. The PIP box allows FAM111A to interact with DNA replication factors, specifically PCNA, whilst the UBL domains play a role in protein–protein interactions. FAM111A is mainly associated with various functions such as antiviral defense [8,9,10], DNA replication [11,12] and genetic disorders [13,14], but its precise role in these diseases is still not fully understood.

FAM111B, a paralog of FAM111A, is a 734 amino acids (85 kDa) protein that exhibits a notable 45% sequence similarity with the FAM111A protein [6]. Before the recent validation of FAM111B protease activity, bioinformatics studies have indicated the likely existence of a trypsin/cysteine protease-like domain located at the C-terminal region of the FAM111B protein [4,7]. Moreover, studies suggest that FAM111B exhibits structural resemblances to FAM111A but notably lacks the PIP box sequence (Figure 1). Instead, it interacts with PCNA through the replication factor C complex (RFC) [4,7]. Furthermore, the FAM111B gene is linked to a reduction in the stability and integrity of the genome, indicating that the gene plays a role in maintaining DNA repair and genome integrity [15]. Consequently, mutations in this specific gene contribute to advancing cancer and fibrosis [16].

Apart from the similarities in the structural and catalytic activity of FAM111A and FAM111B, there appears to be an overlap in their cellular function, as indicated in previous studies [4,6]. For instance, it was previously shown that mutations in both genes hyperactivated their intrinsic protease activity, which triggered replication and transcription shutdown, disruption of microtubule networks and apoptosis in cells. However, both genes in their wild-type forms and physiological levels are not essential in these cellular processes [4]. Moreover, co-immunoprecipitation studies carried out in recent studies validated their connectedness [4,6]. Nonetheless, the implication of these genes in different genetic disorders diseases suggests that each disease participates in other unique pathways that lead to these diseases and warrants studying these genes individually or collectively in each context or overlapping clinical phenotypes to understand the molecular pathophysiology brought about by their dysregulation and potential implications for therapeutic interventions.

## 2. The Biological Role of FAM111A and FAM111B in Cellular Processes

### 2.1. FAM111A at the Replication Fork

Proteomics research suggests that FAM111A is enriched on newly synthesized DNA. FAM111A employs its PIP box to target replication forks by interacting with PCNA [11,12]. PCNA acts as a “sliding clamp” that encircles the DNA double helix at replication forks. This encirclement helps DNA polymerase enzymes in place for synthesizing new DNA strands while attached to the DNA template during replication (Figure 2a). FAM111A also protects replication forks from stalling at protein obstacles caused by nucleoprotein complexes or DNA–protein crosslinks (DPCs) on replicating cells’ DNA [12] (Figure 2b). A study conducted by Kojima et al. (2020) revealed that the levels of specific DPCs: topoisomerase 1 cleavage complexes (TOP1ccs) and poly (ADP-ribose) polymerase 1 (PARP1) DNA complexes (i.e., PARP1-DNA), increased in FAM111A knockout (FAM111A KO) cells [5]. The increase in these DPCs led to the increased sensitivity of FAM111A KO cells to TOP1 and PARP1 inhibitors camptothecin and niraparib, respectively, compared to the wild-type, providing evidence for FAM111A’s important role in genomic stability during replication. More importantly, the same study showed that cells expressing an active site mutant form of FAM111A (i.e., S541A) failed to rescue the replication defects caused by camptothecin and niraparib, suggesting that the protease activity of FAM111A is critical for eliminating these DPCs.

### 2.2. FAM111A as a Viral Replication Restriction Factor

FAM111A, a nuclear protein primarily linked to the regulation of host DNA replication, assumes a significant function as an antiviral agent [8,9]. Notably, it exhibits antiviral activities against DNA and RNA viruses [10,17,18], shedding light on its broad-spectrum antiviral potential. In the context of DNA viruses, such as the vaccinia virus (VACV), two studies showed that FAM111A prevented VACV replication by targeting its DNA-binding protein, I3, for autophagic degradation [9,19]. This antiviral activity was subsequently shown to be counteracted by the serine protease inhibitor 1 (SPI-1), which is conserved across all orthoviruses [9,19], suggesting that FAM111A’s antiviral properties resulted from its proteolytic activity. Moreover, the study by Zhu et al. showed that FAM111A translocated into the cytoplasm upon VACV infection by degrading the nuclear pore complex through its protease activity (Figure 2c, top panel) [9]. FAM111A was also shown to exert its influence on the infection of SV40, also known as Simian Virus 40 (SV40), by serving as a factor that restricts the range of hosts. The viral replication process heavily relies on the presence of the SV40 large T antigen (LT), particularly its C-terminal region, which is deemed crucial for lytic infection in specific cell types [18]. In-depth examination of gene expression and mass spectrometry revealed a distinct interaction between the C-terminal region of LT and FAM111A, and depletion of FAM111A led to heightened viral gene expression and lytic infection of SV40 host range mutants, as well as an escalated replication of adenovirus in cells with restrictive properties [10]. Similarly, the antiviral activity of FAM111A in RNA viruses was demonstrated in Zika virus (ZIKV)-infected A549 cells [17]. FAM111A contributed to the inhibition of ZIKV replication through its overexpression, which was induced by the activation of interferon regulatory factor 2 (IRF2), which, in turn, enhanced the host restriction activity of replication factor C subunit 3 (RFC3) to ultimately prevent viral replication, indicating the potential broader applicability of FAM111A-dependent antiviral mechanisms across virus families (Figure 2c, low panel) [17].

These findings from this study and similar studies underscore the versatile and critical role of FAM111A as a viral host restriction factor predominantly through its trypsin-like domain and partly its DNA-binding domain, which enables it to bind and interact with viral proteins. Therefore, studying FAM111A’s interactions within a broader range of viruses holds promise for uncovering additional facets of its antiviral function and the potential development of therapeutic strategies targeting these interactions.

### 2.3. FAM111B, an Integral Player in Regulating DNA Repair or Replication, Cell Cycle and Apoptosis Regulation

FAM111B gene mutations have also been reported to affect genome stability and integrity adversely [15], contributing to the growing evidence that FAM111B plays a role in maintaining DNA repair and genome integrity. In response to the formation of DPCs or apparent DNA damage, the transcription factor protein p53 activates specific genes that lead to either cellular apoptosis or cell cycle arrest [20,21]. Furthermore, the direct or indirect activation of FAM111B by p53 has also been reported [22]. Moreover, it has been previously shown that the knockout or silencing of FAM111B affects the expression of p53 and p53-related proteins, such as p16, BAG3, BCL-2, CCNB1 [22,23,24], AKT and caspase-1 [23]. Therefore, in addition to FAM111B’s role in DNA repair, its proposed cellular function can be extended to cell cycle regulation.

Initial evidence indicating that FAM111B plays a role in the cell cycle came from a multiomics study that showed a gradual rise in FAM111B RNA levels throughout the G1 phase, ultimately resulting in the noticeable accumulation of the FAM111B protein during the S phase [21]. These findings were validated in a lung adenocarcinoma (LUAD) cell line where FAM111B had been deleted [25]. Notably, there was a marked decrease in the number of cells progressing through the S and G2/M phases of the cell cycle, and a higher proportion of cells remained in the G0/G1 phases. FAM111B was also shown to influence the degradation of p16 (CDKN2A), a protein responsible for delaying the G1/S transition by inhibiting the formation of the CDK 4/6—Cyclin D complex [24,26]. This complex is responsible for phosphorylating the retinoblastoma protein (pRb1), allowing for its dissociation from E2F and promoting cell cycle progression. If the DNA repair process is effective, FAM111B could initiate the resumption of the cell cycle by decreasing the levels of p16 (Figure 3a) [16,26].

In alignment with the previously proposed and experimentally determined function of FAM111B, a recent study showed that the knockout of FAM111B caused an increase in p16, E2F and pRb1 expression levels [22] and concluded that FAM111B further regulates the cell cycle by controlling the activity of cyclin B1 and CDC25C. The protein cyclin B1 regulates the cell cycle passage from the G2 to the M phase, which is necessary for healthy cell division. Cyclin B1 binds to and activates CDK1, which sets off a series of processes that lead to mitosis. Activated p53 and, possibly, FAM111B inhibit cyclin B1, thereby delaying the G2/M transition and allowing more time for DNA repair [16] (Figure 3).

The function of the FAM111B protein in the S phase of the cell cycle was elucidated by discovering its interaction with DNA-binding proteins like replication factor C subunit 1 (RFC1) and proliferating cell nuclear antigen (PCNA) [4]. RFC is a complex comprising five subunits vital in loading PCNA onto DNA, a critical step in DNA replication [27]. The disruption of PCNA and RFC was reported in cells overexpressing FAM111B or cells expressing mutant FAM111B derived from patients [4], supporting its role in DNA replication.

FAM111B was suggested to influence the apoptotic pathway by upregulating the expression of anti-apoptotic proteins BCL2 and BAG3 (Bcl-2-associated athanogene3) [22]. BCL2 inhibits the release of cytochrome c into the cytoplasm whilst BAG3 simultaneously interacts with and inhibits the pro-apoptotic protein BAX; in this way, programmed cell death is suppressed, resulting in cell survival [28,29]. Furthermore, Sun et al. (2019) showed that FAM111B may bind directly with BAG3 by demonstrating that the expression of FAM111B was decreased with a decreased BAG3 and BCL2 expression [22]. All these findings provide strong evidence for FAM111B in DNA damage repair and replication, cell division and apoptosis and warrant further studies in this context.

### 2.4. The Involvement of FAM111B in Nuclear Transport and Telomere Length Maintenance

Studies have reported on the mainly nuclear localization of FAM111B [4,30] and thus proposed nucleo-specific roles as previously described [16]. A recent study on the cellular function of FAM111B showed it interacted with specific components of the nuclear pore complexes, precisely two nucleoporins, SEC13 and NUP42 [31]. This study indicated that FAM111B is recruited to the nuclear periphery through this interaction to perform its function [31]. Furthermore, FAM111B’s association with these nucleoporins suggests its involvement in nucleo-cytoplasmic trafficking, specifically in mRNA exports. However, the study reported no significant changes in the global mRNA transport in wild-type and FAM111B KO cells [31]. This study, however, showed by mass spectroscopy that FAM111B interacts with the ribonucleic acid export 1 (RAE1) and GLE1 RNA export mediator (GLE1) proteins, which are components of the RNA export machinery. However, the exact mechanisms remain unclear and warrant further studies to validate this role.

Kliszczak et al. (2023) also showed that FAM111B is essential in maintaining normal telomere length [31]. In FAM111B KO cells, telomeres were reported to be shorter due to the lower recruitment of the telomere repeat binding factor 2 (TRF2) component of shelterin (or telosome, a protein that protects telomeres and regulates telomerase activity) [32]. The loss of TRF2 or shelterin has been associated with genome instability and the activation of DNA damage response and repair mechanisms in cells, including end-to-end fusion [33], non-homology or homologous directed repair [34,35], p53 activation and ATM-mediated apoptosis [36]. In other words, the loss of FAM111B or disease-associated mutation results in the loss of TRF2, leading to the critical shortening of telomeres and genome instability, further providing evidence for FAM111B in this vein.

### 2.5. The Overlap in the Cellular Functions of FAM111A and FAM111B

The cellular functions of FAM111A and FAM111B exhibit notable overlap, particularly in their roles related to DNA replication, transcription regulation, microtubule organization and apoptotic pathways [4]. FAM111A dominant missense mutations that hyperactivate its intrinsic protease activity have been shown to disrupt DNA replication, transcription and apoptosis induction [4]. Similarly, FAM111B heterozygous missense mutations near its protease domain have been demonstrated to elevate its protease activity and subsequent impairment of DNA synthesis, transcriptional processes, microtubule integrity and apoptotic cell death [4,30].

Moreover, FAM111A and FAM111B proteins have been shown to interact with cellular replication and transcription components like RFC subunits and RPB1, suggesting potential functional synergy [4]. Furthermore, a FAM111B-enriched interactome analysis in another study identified FAM111B as a primary interactor of FAM111A [6], supporting their collaborative roles, possibly by forming a joint complex to regulate various cellular processes. Therefore, despite the differences in the clinical manifestations of disorders associated with FAM111A and FAM111B mutations, their pathological mechanisms converge on the dysregulation of protease activity, highlighting a shared molecular basis underlying their disease-promoting potential.

## 3. Dysregulation of FAM111 Proteases in Genetic Diseases, including Cancer-Associated Fibrosing Disease

### 3.1. FAM111A Mutations in Genetic Diseases and Cancer

Mutations within the FAM111A gene have been identified as the genetic basis for two related disorders: Kenny–Caffey Syndrome type 2 (KCS2) and the more severe Gracile Bone Dysplasia (GCLEB), also known as osteocraniostenosis (OCS) [7,14,36]. In both conditions, affected individuals exhibit symptoms such as narrowed and thickened long bones, hypoparathyroidism, low levels of calcium in the blood (hypocalcemia) and short stature. Currently, 35 cases of KCS2, GLEB and newly described clinical syndromes with associated FAM111A mutations are reported (Table 1a). These are typically missense mutations, and they cluster in two specific regions of the FAM111A protein (reviewed in [7]). One cluster lies within the trypsin 2 enzyme domain, while the second is between the UBLs (ubiquitin-like domains) and the Trypsin 2 domain (Figure 1). Studies have revealed that most FAM111A and FAM111B variants associated with patients are gain-of-function mutations, leading to overactivity in the resultant protein [4,5]. When patient-associated FAM111A mutants are overexpressed, they cause various cellular issues, including reduced DNA synthesis, disruption of nuclear structure and cell death [8] (Figure 4), suggesting that one potential mechanism underlying these diseases is the degradation of essential proteins by dysregulated FAM111A. However, a recent review argued against this model because it does not explain cases where the active site serine in FAM111A is mutated, which should theoretically inactivate the enzyme, and it suggested that both hyperactivation and inactivation of FAM111A can lead to KCS2 [7]. Specifically, the hyperactive mutants of FAM111A may degrade the wild-type FAM111A protein, depleting it, while inactivating mutants may trap and sequester substrates from the wild-type FAM111A protein, interfering with its normal function. Therefore, future studies are required to validate these hypotheses. Recently, long noncoding RNA (lncRNA) of FAM111A (known as FAM111A-divergent transcript or FAM111A-DT) have been shown to promote the progression and poor survival rate of hepatocellular carcinoma (HCC) through its modification by N^6^-methyladenosine (m^6^A) [37]. The m6A-modified FAM111DT was shown to drive the overexpression of wild-type FAM111A by activating the FAM111A promoter through interaction with the m6A-binding protein YTHDC1 and the recruitment of the histone demethylase KDM3B to FAM111A promoter. Furthermore, this study showed that m6A-modified FAM111A-DT promoted HCC cellular proliferation, DNA replication and tumor growth, and mutation of m6A sites on FAM111A-DT abolished these roles [37] and could represent a viable treatment target for HCC. Additionally, other studies have associated FAM111A variants with prostate cancer predisposition [38], a novel prognostic and immunologic biomarker for diffuse lower-grade glioma [39] and early diagnostic indicator of gastric cancer [40], thus highlighting its essential role in the diagnosis and treatment of cancers.

### 3.2. FAM111B and Fibrosis

Mutations in the FAM111B gene are linked to a rare hereditary fibrosing poikiloderma syndrome characterized by poikiloderma, tendon contractures, muscle weakness and pulmonary fibrosis, collectively known as POIKTMP [41]. There are 39 cases of POIKTMP with associated FAM111B mutations (Table 1b) to date. A recent review has attempted to provide insights into the molecular basis of the variants in FAM111B in the context of its gain-of-function and accelerated autocleavage and other proteins crucial for cell homeostasis [16]. This review, authored by the same team, also stated that fibrosis, marked by excessive deposition of extracellular matrix components in tissues and organs, occurs due to fibroblasts failing to undergo apoptosis and indicated that this failure could result from the negative feedback from the increasing apoptosis occurring in the adjoining epithelial cells during tissue insult.

A recent study on the cellular function of FAM111B, apart from corroborating this proposed function, shed more light on its role in fibrosis [31]. This study showed that in FAM111B-deficient cells and cells expressing a POIKTMP-associated FAM111B mutation, many chromosomes exhibited the absence of telomeric DNA and underwent fusion events, thus indicating that the lack of FAM111B resulted in the development of shortened telomeres, which could eventually lead to genome instability (Figure 4). This observation aligns with previously reported spurious chromosomal abnormalities related to FAM111B mutations in patients with POIKTMP [15]. Moreover, several studies have associated telomere shortening with pulmonary dysfunction and fibrosis [42,43,44,45], which is a major life-threatening clinical presentation in POIKTMP patients. However, normal telomere length was reported for one patient [46]; therefore, telomere shortening should be screened or validated in patients.

### 3.3. FAM111B and Cancer Progression

The overexpression of the FAM111B gene is linked to various cancers, including lung adenocarcinoma [22], fibrosarcoma [30], and breast [47] and ovarian [48] cancers. Cancer and non-cancer-associated FAM111B mutations have also been reported (reviewed in [30]). This study, which also involved the in-silico analysis of FAM111B RNA expression in cancerous and non-cancer cell lines, showed that 68% of cancerous and 32% of non-cancer cell lines significantly expressed this gene [30]. Moreover, an extensive pan-cancer analysis of FAM111B using datasets from cancer and gene expression databases and clinical cohorts of gastric cancer patients suggested including FAM111B as a prognostic biomarker for various cancers [48]. A plausible explanation for how FAM111B overexpression supports cancer progression is that it may influence the apoptotic pathway by upregulating the expression of the anti-apoptotic proteins Bcl-2 and BAG3 or cause indiscriminate degradation of other DNA-associated proteins, including replication or transcription factors (like RFC1 and RPB1), histones and cell-cycle-related proteins such as p16 [4,7,16].

Additionally, a recent gene-enrichment analysis carried out on The Cancer Genome Atlas (TCGA) datasets of various cancers suggested that FAM111B promotes cancer via the dysregulation of the immune process, chromosome stability and DNA repair and, in more mechanistic terms, facilitates the growth of malignant tumor cells by preventing apoptosis [48]. Similarly, the study by Rhoda et al. (2023), apart from confirming a robust nuclear localization of FAM111B, showed the co-localization with centrosomes and mitotic spindles during cell division, suggesting that FAM111B plays a role in regulating cellular growth, movement and critical signaling processes that influence fundamental cellular functions [30]. Certain cancers have been associated with microtubule instability, and this insight may aid in comprehending the involvement of FAM111B in processes like cell migration, proliferation and apoptosis [49,50]. Moreover, the heightened expression of FAM111B in cancer could potentially impact the dynamics of spindle microtubules, a crucial component in cell division. This observation was also supported by another study that showed that FAM111B contributes to tumor growth and metastasis through interaction with the transforming acidic coiled-coil protein 3 (TCC3), which, in turn, activates the PI3/AKT pathway in hepatocellular carcinoma [51]. An essential role of TCC3 is maintaining the spindle and centrosome localization through phosphorylation by Aurora A kinase in mitosis [52].

**Table 1 ijms-25-02845-t001:** The clinical cases of (**a**) *FAM111A* and (**b**) *FAM111B* gene variants/mutations and associated diseases.

(a)			
Case No.	*FAM111A* Variants	Disease	Reference(s)
1	c.968G > A (p. Gly323Glu)	Hypoparathyroidism	[53]
2	c.968G > A (p. Gly323Glu)	KCS2	[54]
3	c.976T > A (p. Leu326Ile) (c. 1714_1716del) and c.1714_1716delATT (p. Ile572del)	KCS2	[55]
4	c.1714_1716delATT (p. Ile572del)	Hypoparathyroidism	[53]
5	c.1012A > G (p. Thr338Ala)	GCLEB	[14]
6–8	c.1026_1028delTTC (p. Ser342del)	GCLEB	[14,56]
9	c.1026_1028delTCG (p. Ser343del)	GCLEB	[57]
10	c.1454G > T (p. Cys485Phe	KCS2	[58]
11	c.1531T > C (p. Tyr511His)	KCS2	[14]
12	c.1542G > T (p. Met514Ile)	GCLEB	[57]
13	c.1579C > A (p. Pro527Thr)	GCLEB	[14]
14	c.1583A > G (p. Asp528Gly)	GCLEB	[14]
15	c.1622C > A (p. Ser541Tyr)	KCS2	[59]
16–17	c.1621T > C (p. Ser541Pro)	KCS2	[60]
18	c.1685A > C (p.Tyr562Ser/Y562S)	KCS2	[61]
19–22	c.1706G > A (p. Arg569His)	KCS2	[14]
23–24	c.1706G > A (p. Arg569His)	KCS2	[13]
25–28	c.1706G > A (p. Arg569His	KCS2	[62]
29	c.1706G > A (p. Arg569His	KCS2	[63]
30	c.1706G > A (p. Arg569His	Hypoparathyroidism	[53]
31	c.1706G > A (p. Arg569His)	KCS2 and Sanjad-Sakati syndrome	[64]
32	c.1706G > A (p. Arg569His)	KCS2	[65]
33	c.1706G > A (p. Arg569His)	Nanophthalmos	[66]
34	c.1706G > A (p. Arg569His)	KCS2	[67]
35	c.1706G > A (p. Arg569His)	KCS2	[68]
**(b)**			
**Case No.**	***FAM111B* Variants**	**Disease**	**Reference(s)**
1,2	c.1247T > C (p. Phe416Ser)	POIKTMP	[69]
3–12	c.1261_1263delAAG (p. Lys421del)	POIKTMP	[46]
13,14	c.1289A > C (p. Gln430Pro)	POIKTMP	[70,71]
15	c.1292T > C (p. Phe431Ser)	Autoimmune polyendocrinopathy-candidiasis-ectodermal dystrophy (APECED)-like/POIKTMP	[72]
16–19	c.1861T > G (p. Tyr621Asp)	POIKTMP	[41]
20	c.1873A > C (p. Thr625Pro)	POIKTMP	[73]
21	c.1874C > A (p. Thr625Asn)	POIKTMP	[71]
22–28	c.1879A > G (p. Arg627Gly)	POIKTMP and pancreatic cancer in case 25	[15,41,71,74,75]
29	c.1881 C > T * (p. Arg627Ser)	POIKTMP	[76]
30–33	c.1883G > A (p. Ser628Asn)	POIKTMP	[41,71,77,78]
34–37	c.1884T > A (p. Ser628Arg)	POIKTMP and pancreatic cancer in case 34	[79]
38	c.1886T > G (p. Phe629Cys)	POIKTMP with end-stage liver disease	[80]
39	c.1886T > C (p. Phe629Ser)	POIKTMP	[81]

* Mutation is not verifiable.

## 4. Conclusions

Based on the published literature, the FAM111 serine proteases FAM111A and FAM111B are intriguing proteins with multifaceted roles in various cellular processes. Proteases are vital in controlling cellular proliferation, differentiation and programmed cell death, making them essential cellular regulation components. The FAM111A protein has been implicated in critical cellular functions. FAM111A’s role in DNA replication is noteworthy, as it assists in loading PCNA onto chromatin, ensuring the stable attachment of DNA polymerases during replication. Moreover, it plays a crucial role in safeguarding replication forks from obstacles such as DNA–protein crosslinks (DPCs), highlighting its significance in preserving DNA stability. Multiple studies have also shown its crucial role as an antiviral factor in defending against various viruses.

On the other hand, FAM111B, a paralog of FAM111A, plays a pivotal role in regulating the cell cycle, DNA repair and apoptosis. It is activated in response to DNA damage, perhaps induced by DPC formation, and orchestrates cell cycle progression and DNA repair processes. FAM111B also modulates apoptosis-related proteins like Bcl-2 and BAG3, impacting cell survival.

However, FAM111A and FAM111B dysregulation has analogous cellular consequences, including DNA replication and transcriptional shutdown, microtubule integrity disruption and apoptosis induction. This functional convergence is underscored in their structural similarities and direct interaction. Additionally, disease-associated mutations in both proteins abolish inhibitory constraints on their protease activities, leading to cellular dysfunction.

*FAM111A* mutations have been implicated in genetic disorders like Kenny–Caffey Syndrome type 2 (KCS2) and Gracile Bone Dysplasia (GCLEB). On the other hand, *FAM111B* mutations are attributed to POIKTMP. In cancer, FAM111B overexpression has been linked to tumor progression, potentially due to its involvement in DNA repair and cell cycle regulation. Interestingly, though FAM111A and FAM111B exhibit a certain degree of interconnectedness and functional similarity, have distinct roles in cellular processes. Therefore, understanding the intricate roles of FAM111A and FAM111B collectively or individually in these cellular processes will shed light on their potential as therapeutic targets and diagnostic markers for various diseases.

In summary, the FAM111 serine proteases FAM111A and FAM111B are emerging as critical players in cellular regulation, DNA repair and disease pathogenesis. Future research is needed to unravel the mechanisms underlying their functions and potential implications for therapeutic interventions in various diseases.

## Figures and Tables

**Figure 1 ijms-25-02845-f001:**
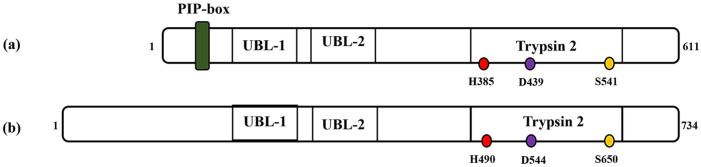
Schematic representations of (**a**) FAM111A and (**b**) FAM111B protein. PIP: PCNA-interacting peptide box; UBL-1: Ubiquitin-like domain 1; UBL-2: Ubiquitin-like domain 2; Trypsin 2: Trypsin-like peptidase domain. Histidine, aspartic acid and serine amino acid (catalytic triads) are depicted in red, purple and yellow, respectively. The numbers indicate the protein length (1–611 for FAM111A and 1–734, FAM111B) and position of the catalytic amino acid residues (i.e., catalytic triads) on each protein.

**Figure 2 ijms-25-02845-f002:**
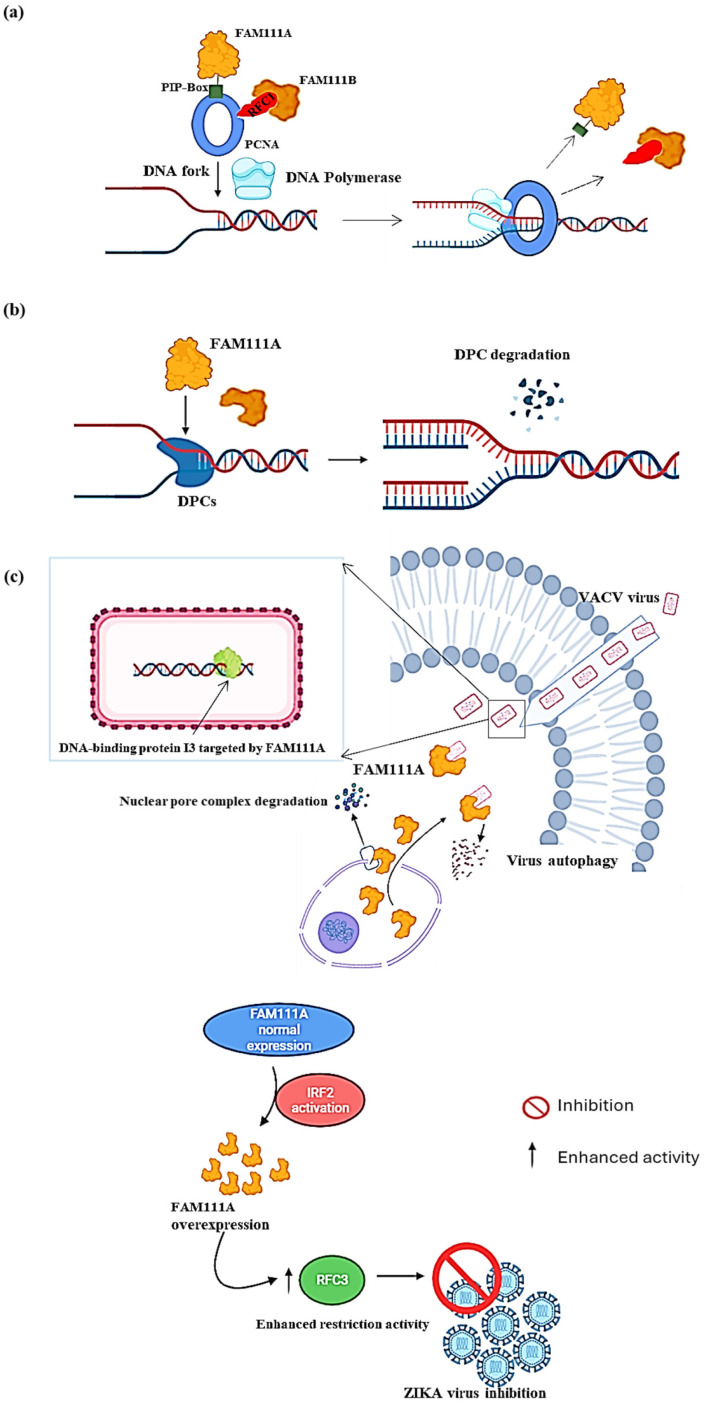
Molecular function of FAM111A. (**a**) At the replication fork (PCNA loading). FAM111A helps load PCNA at the replication fork using its PIP box. Once the PCNA is loaded, DNA polymerase can firmly adhere to the DNA strand and progress with DNA replication. (**b**) At the replication fork (DPCs degradation). FAM111A targets DNA–protein crosslinks (DPCs) to prevent stalling the replication process or causing DNA breaks. After locating the DPCs, FAM111A uses its proteolytic abilities to degrade them, which allows for the progression of DNA replication. (**c**) As a viral replication restriction factor. In the DNA virus VACV (**top panel**), FAM111A translocates into the cytoplasm by degrading the nuclear pore complex of the cell by its protease activity and then targets the virus DNA-binding protein I3 for degradation via autophagy. In the RNA virus ZIKA (**bottom panel**), FAM111A overexpression is induced by the activation of interferon regulatory factor 2 (IRF2), leading to replication factor C subunit 3 (RFC3) activity enhancement in host cells, thereby increasing host viral restriction activity and ultimately inhibiting ZIKA virus replication. Image created with Biorender (https://app.biorender.com/illustrations/626a882bceb6a1798ccc52cc, accessed on 3 January 2024).

**Figure 3 ijms-25-02845-f003:**
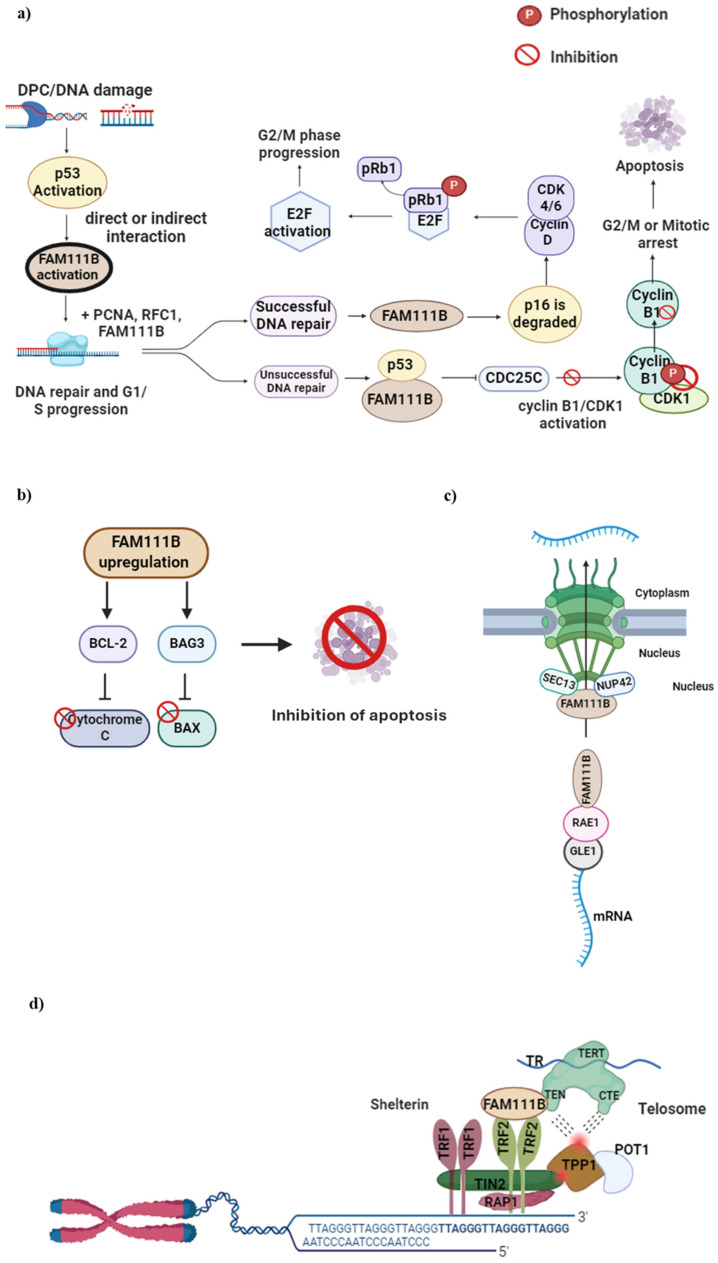
FAM111B functions in DNA repair, cell cycle, apoptosis, nuclear transport and telomere maintenance. (**a**) DPC formation or DNA damage in the G1 to S phase activates the p53 protein, which, in turn, activates FAM111B. Upon successful DNA repair, FAM111B degrades p16, leading to CDK 4/6—Cyclin D complex formation. This complex phosphorylates pRb1 and dissociates it from E2F, which then becomes activated, allowing cell cycle progression to G2/M. When DNA repair is unsuccessful, FAM111B and p53 inhibit C Cyclin B, resulting in delayed cell cycle progression into the G2/M phase. (**b**) FAM111B upregulation activates BCl-2 and BAG3, inhibiting pro-apoptotic Cytochrome C and BAX proteins, respectively, preventing cellular apoptosis and cell survival, suggesting FAM111B is involved in apoptosis. (**c**) FAM111B may function as a nuclear export protein through interactions with nucleoporins SEC13 and NUP42 and mRNA export via binding to the RNA export proteins RAE1 and GLE1 RNA. (**d**) FAM111B functions in telomerase maintenance by recruiting TRF2, a shelterin protein complex component vital for healthy telomeres. Image created with Biorender (https://app.biorender.com/illustrations/626a882bceb6a1798ccc52cc, accessed on 7 January 2024).

**Figure 4 ijms-25-02845-f004:**
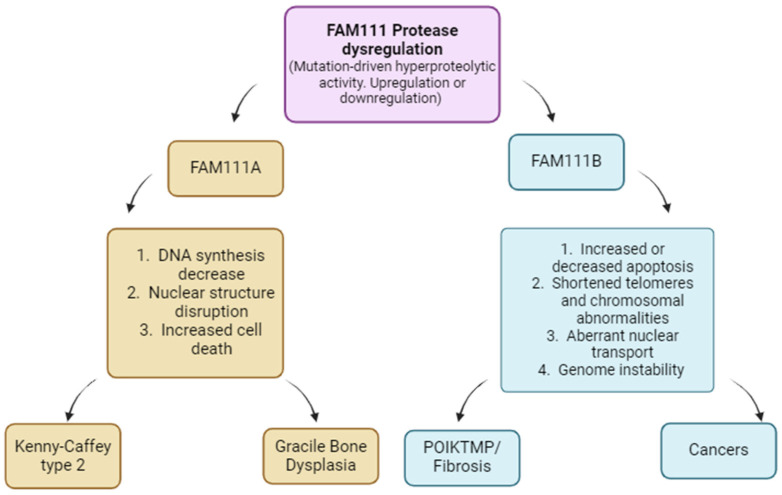
Disrupted cellular processes resulting from FAM111A and FAM111B dysregulation and associated diseases. Image created with Biorender (https://app.biorender.com/illustrations/626a882bceb6a17ccc52cc, accessed on 15 January 2024).

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
