# Peer review of "Unravelling the Intricate Roles of FAM111A and FAM111B: From Protease-Mediated Cellular Processes to Disease Implications"

_ijms, 2024, doi:10.3390/ijms25052845_

Round 1

Reviewer 1 Report

Comments and Suggestions for Authors

The manuscript is well-written overall and provides a comprehensive review of FAM111A and FAM111B's roles in various cellular processes, including viral defense, tumorigenesis, and DNA replication modulation. Welter et al. published a similar high-quality review in 2022. As the result, this manuscript could benefit from focusing more on studies published in 2023 and further expand its reference list. Additionally, the quality of Figure 3 needs enhancement, as its current format compromises readability.

Additional concerns are as follows:

- At Line 106, appropriate citations need to be inserted for accuracy.

- In Line 98, it's noted that there are two studies on FAM111A's antiviral activity against VACV. The study by Panda et al. should also be mentioned here.

- Further, the manuscript should include references to reports on FAM111A's antiviral function against SV40 and its role in regulating the SV40 host range, as detailed in Tarnita et al. (2018) and Fine et al. (2012). These references are particularly relevant for Section 2.2.

Regarding the figures:

- In Figure 2c, I recommend revising the diagram depicting VACV, as poxviruses do not have “spikes”. Additionally, the lower part of Figure 2c seems to align more with the results published by Panda et al., using VACV as the model virus.

- At Line 194, replace "K.O." with "KO" for consistency in abbreviations.

- At Line 219, "FAM111" should be corrected to "FAM111A" for specificity.

- The resolution of Figure 3 requires improvement. Currently, the figure is blurred, making some words in Figure 3c unreadable.

Comments on the Quality of English Language

N/A

Author Response

Comments and Suggestions for Authors

The manuscript is well-written overall and provides a comprehensive review of FAM111A and FAM111B's roles in various cellular processes, including viral defense, tumorigenesis, and DNA replication modulation. Welter et al. published a similar high-quality review in 2022. As the result, this manuscript could benefit from focusing more on studies published in 2023 and further expand its reference list. Additionally, the quality of Figure 3 needs enhancement, as its current format compromises readability.

Response: We like to use this medium to appreciate the reviewer for their kind words and valuable feedback on our manuscript. Below are the responses to the comments in the first paragraph.

We have improved and expanded the manuscript by providing the latest studies, including studies carried out in our research group (the additions are highlighted in yellow in the manuscript version). We have reworked the image and provided a better resolution.

 Additional concerns are as follows:

- At Line 106, appropriate citations need to be inserted for accuracy.

Response: We have addressed this point (now lines 127 to 130)

- In Line 98, it's noted that there are two studies on FAM111A's antiviral activity against VACV. The study by Panda et al. should also be mentioned here.

Response: We have included this reference (lines 112 and 114)

- Further, the manuscript should include references to reports on FAM111A's antiviral function against SV40 and its role in regulating the SV40 host range, as detailed in Tarnita et al. (2018) and Fine et al. (2012). These references are particularly relevant for Section 2.2.

Response: Fine et al. was already included in the previous version and have included Tarnital et al. and added a section on the SV40 host range (lines 110  to 125) in the new version.

Regarding the figures:

- In Figure 2c, I recommend revising the diagram depicting VACV, as poxviruses do not have "spikes". Additionally, the lower part of Figure 2c seems to align more with the results published by Panda et al., using VACV as the model virus.

Response: We revised the diagram as pointed out. The idea behind the image is to illustrate FAM111A as a restriction factor of DNA and RNA replication.

- At Line 194, replace "K.O." with "KO" for consistency in abbreviations.

Response: Corrected (line 212). Thanks.

- At Line 219, "FAM111" should be corrected to "FAM111A" for specificity.

- The resolution of Figure 3 requires improvement. Currently, the figure is blurred, making some words in Figure 3c unreadable.

Response: We have corrected this error (line 274). We have fixed the resolution of this figure in the new version.

Reviewer 2 Report

Comments and Suggestions for Authors

The authors propose a review entitled "Unravelling the intricate roles of FAM111A and FAM111B: From protease-mediated cellular processes to disease implications". The roles of FAM111A and FAM111B are still poorly characterized in the literature, which makes this review interesting.

In terms of form, the manuscript is not very easy to read, with a tendency towards a catalog style; the conclusions are not very clear and there are numerous typographical and presentation errors in the figures.

For example:

-          p4; figure 2: pink frames are visible in the background.

-          p5; line 183: “FAM111B » repeated twice in the same sentence

-          p7; figure 3: “FAM11B” instead of “FAM111B”

-          p8; 3.3) all references appear in the text without brackets.

-          p9; figure “3” is figure 4; “activity” repeated twice;

-          reference 30 = reference 37

1) The review could be more substantial, particularly for FAM111A. It is clear that the authors are working more specifically on the FAM111B protein and that the FAM111A part is less developed.

2) Regarding pathologies caused by pathogenic variants in FAM111A and FAM111B, this section could be further developed in the review with more extensive references, including publications of the largest cohorts.

p6, line 211: reference 14 should be cited too.

3) The manuscript lacks rigor: it should be noted which is the source publication and not a citation from another review or self-citation.

The authors rely on a "recent review" (reference 17; Arowolo et al, 2022) which is a review by the same team but does not specify that it is their work.

Other example, the direct interaction between p53 and FAM111B (p3, line 123) has not been documented in the literature. The authors cite reference 21 (Kawasaki et al, 2020) who themselves in this publication cite Sun et al, 2019; but Sun et al's references to carry this claim are not scientifically proven. This point needs to be reviewed throughout the manuscript.

4) On the other hand, the review includes many suggestions about the role of these proteins that either need to be better argued or withdrawn.  Figure 3 is unclear in its reading and includes too many assumptions to be published.

5) As the title suggests, we would expect to see in-depth work on the interconnected role of FAM111A and FAM111B, which is what made this review so interesting.  The manuscript is very disappointing in this respect, providing insufficient information on the interaction between FAM111A and FAM111B which is not mentioned or discussed in this review despite having been published (Hoffmann et al, 2020).

6) Concerning the decrease in telomere length (p9, lines 258-267), it should be added that normal telomere length was reported for one patient (Seo et al, 2016). Further studies are therefore needed to affirm a decrease in telomere length in patients.

Author Response

Comments and Suggestions for Authors

The authors propose a review entitled "Unravelling the intricate roles of FAM111A and FAM111B: From protease-mediated cellular processes to disease implications". The roles of FAM111A and FAM111B are still poorly characterized in the literature, which makes this review interesting.

In terms of form, the manuscript is not very easy to read, with a tendency towards a catalog style; the conclusions are not very clear and there are numerous typographical and presentation errors in the figures.

Response: We appreciate your valuable feedback on the manuscript and are sorry you found the manuscript difficult to read. Consequently, we have done our best to improve the readability of the current version as per the suggestions. The corrections have also been highlighted in yellow.

For example:

-          p4; figure 2: pink frames are visible in the background.

Response: We have addressed this point in the current version.

-          p5; line 183: "FAM111B » repeated twice in the same sentence

Response: We have corrected the statement (now lines 205 to 206).

-          p7; figure 3: “FAM11B” instead of “FAM111B”

Response: Corrected, thank you.

-          p8; 3.3) all references appear in the text without brackets.

Response: Corrected, thank you.

-          p9; figure “3” is figure 4; “activity” repeated twice;

Response: Figure numbering corrected. About the alleged error of repeating "activity" twice, we can't find this error on the old or new draft.

-          reference 30 = reference 37

Response: Corrected, thank you.

  • The review could be more substantial, particularly for FAM111A. It is clear that the authors are working more specifically on the FAM111B protein and that the FAM111A part is less developed.

Response: Addressed in the latest version, changes are highlighted in yellow.

2) Regarding pathologies caused by pathogenic variants in FAM111A and FAM111B, this section could be further developed in the review with more extensive references, including publications of the largest cohorts.

Response: In the current version, we have provided an updated record of the cases and associated FAM111A and FAM111B (lines 269 and 302, Table 1a and b).

p6, line 211: reference 14 should be cited too.

Response: Reference cited (line 266 in the current version)

3) The manuscript lacks rigor: it should be noted which is the source publication and not a citation from another review or self-citation.

The authors rely on a "recent review" (reference 17; Arowolo et al, 2022) which is a review by the same team but does not specify that it is their work.

Response: This point has been addressed in the current version.

Other example, the direct interaction between p53 and FAM111B (p3, line 123) has not been documented in the literature. The authors cite reference 21 (Kawasaki et al, 2020) who themselves in this publication cite Sun et al, 2019; but Sun et al's references to carry this claim are not scientifically proven. This point needs to be reviewed throughout the manuscript.

Response: We have addressed this point in the current version and provided supporting references (lines 160  to 165).

4) On the other hand, the review includes many suggestions about the role of these proteins that either need to be better argued or withdrawn.  Figure 3 is unclear in its reading and includes too many assumptions to be published.

Response: We have corrected this image, which we believe is a vital component to include in the manuscript as it can aid interested researchers in identifying areas of future research on the topic.

5) As the title suggests, we would expect to see in-depth work on the interconnected role of FAM111A and FAM111B, which is what made this review so interesting.  The manuscript is very disappointing in this respect, providing insufficient information on the interaction between FAM111A and FAM111B which is not mentioned or discussed in this review despite having been published (Hoffmann et al, 2020).

Response: The main objective of this review is to highlight what is currently known about the cellular function of FAM111A and FAM111B (which remains widely uncharacterized) and their potential use as diagnostic or therapeutic targets in diseases. Nonetheless, we have included a brief section to address the reviewer's point. (Section 2.5) and included a statement in the introduction (lines 69 -80) and conclusion (line 397).

6) Concerning the decrease in telomere length (p9, lines 258-267), it should be added that normal telomere length was reported for one patient (Seo et al, 2016). Further studies are therefore needed to affirm a decrease in telomere length in patients.

Response: Noted with thanks, we have included this statement and cited the reference (line 321).

Round 2

Reviewer 2 Report

Comments and Suggestions for Authors

We appreciate the authors for submitting this revised manuscript. However, some changes still need to be made or have not been implemented following our comments.

- Throughout the text, replace the term "mutations" with "variants" or "pathogenic variants."

- p5, line 162: The subject of the sentence is missing.

- p6, line 207: Reference 31 should be cited at the end of the sentence below and not the following one: "A recent study on the cellular function of FAM111B showed it interacted with specific components of the nuclear pore complexes, precisely two nucleoporins, SEC13 and NUP42."

- Figure 3: Part d) lacks readability; the font size is generally too small, and the text on a dark purple background is unreadable.

- p8, line 274: Modify the sentence as follows: "Studies have revealed that most FAM111A variants, as well as FAM111B variants, associated with patients are gain-of-function variants, leading to overactivity in the resultant protein."

- p9, lines 305-311: The requested modification has not been implemented. This is a self-citation since the recent review [16] is authored by the same authors. It should be clearly noted in the text that this is the same team.

"A recent review has attempted to provide insights into the molecular basis of the variants in FAM111B in the context of its gain-of-function and accelerated autocleavage and other proteins crucial for cell homeostasis [16]. This review, authored by the same team, also stated that fibrosis, marked by excessive deposition of extracellular matrix components in tissues and organs, occurs due to fibroblasts failing to undergo apoptosis and indicated that this failure could result from the negative feedback from the increasing apoptosis occurring in the adjoining epithelial cells during tissue insult."

- We suggest that Table 1 be placed in supplementary data, and the formatting should be revised for a more readable presentation, particularly with two columns for nucleotide variants and protein variants.

- Table 1:

  • Cases 13, 14: References 70 and 73 (not 46)

  • Cases 22-26 (not 27): References 15, 41, 73, 74, 75; only one new case published by Roversi

  • Case 28 => to be modified to Case 27: Nomenclature error: p.(Arg627Ser)

  • Cases 28-32 (instead of 29-33)

  • Cases 33-36 (instead of 34-37): Nomenclature error p.(Ser628Asn)

  • Case 37 (instead of 38): A missing case: Case 38, published by Hoëger et al., 2023

- Figure 4: The repetition "activity activity" has not been corrected. We suggest separating the effects of pathogenic variants responsible for POIKTMP/Fibrosis from the effects responsible for cancers.

Author Response

Dear Editor/Reviewer 2,

We are most grateful for the kind opportunity to revise our manuscript for a second time and for the valuable corrections and feedback. Please see below the response to the comments/suggestions. The changes made are highlighted in yellow.

Best regards,

We appreciate the authors for submitting this revised manuscript. However, some changes still need to be made or have not been implemented following our comments.

- Throughout the text, replace the term "mutations" with "variants" or "pathogenic variants."

Response: Thanks for the suggestion. We believe, however, that the words "mutation", "variant", and "pathogenic variant" can be used interchangeably in this manuscript's context (this is not a population-based study for which the point raised by the reviewer might hold).

- p5, line 162: The subject of the sentence is missing.

Response: We have corrected this point (text highlighted in red and correction highlighted in yellow).

- p6, line 207: Reference 31 should be cited at the end of the sentence below and not the following one: "A recent study on the cellular function of FAM111B showed it interacted with specific components of the nuclear pore complexes, precisely two nucleoporins, SEC13 and NUP42."

Response: We have included this reference.

- Figure 3: Part d) lacks readability; the font size is generally too small, and the text on a dark purple background is unreadable

Response: We have increased the font size and lightened the purple colour to aid readability.

- p8, line 274: Modify the sentence as follows: "Studies have revealed that most FAM111A variants, as well as FAM111B variants, associated with patients are gain-of-function variants, leading to overactivity in the resultant protein."

Response: We have addressed this point.

- p9, lines 305-311: The requested modification has not been implemented. This is a self-citation since the recent review [16] is authored by the same authors. It should be clearly noted in the text that this is the same team.

"A recent review has attempted to provide insights into the molecular basis of the variants in FAM111B in the context of its gain-of-function and accelerated autocleavage and other proteins crucial for cell homeostasis [16]. This review, authored by the same team, also stated that fibrosis, marked by excessive deposition of extracellular matrix components in tissues and organs, occurs due to fibroblasts failing to undergo apoptosis and indicated that this failure could result from the negative feedback from the increasing apoptosis occurring in the adjoining epithelial cells during tissue insult."-

Response: We have done exactly that, thank you.

- We suggest that Table 1 be placed in supplementary data, and the formatting should be revised for a more readable presentation, particularly with two columns for nucleotide variants and protein variants.

Response: We have revised the table format, and we think the information column in its current form provides the necessary information. Given the valuable insights and corrections provided by the reviewer, we believe it would be a shame to put the Table as supplementary information. Nonetheless, we are open to this suggestion if the journal editors deem it fit.

- Table 1:

  • Cases 13, 14: References 70 and 73 (not 46) –

Response: Error corrected, thank you.

  • Cases 22-26 (not 27): References 15, 41, 73, 74, 75; only one new case published by Roversi

Response: 2 cases (two unrelated Italians) were in fact published by Roversi (ref. 15), 3 cases by Mercier et al in 2013 (Ref. 41, one of these cases was re-mentioned in Ref.74), 1 addition by the same group in 2015 (in Ref. in 73) and 1 last case by ref. 75. Therefore, the total number of cases with this mutation/variant is 7 cases (22-28)

  • Case 28 => to be modified to Case 27: Nomenclature error: p.(Arg627Ser)

Response: This case number is now Case 29, and variant nomenclature has been corrected. However, the variant is not verifiable because the nucleotide at the reported nucleotide position (i.e., 1881) is A and not C; therefore, the C>T variant is incorrect. A letter to the editor will follow suit to correct this error in this article.

  • Cases 28-32 (instead of 29-33)

Response:  This will now be 30-33

  • Cases 33-36 (instead of 34-37): Nomenclature error p. (Ser628Asn)

Response:  This Case number remains unchanged. We have rechecked, and there is no nomenclature error for this mutation. The mutation is p.Ser628Arg, not p. Ser628Asn as seen in cases 30-33.

  • Case 37 (instead of 38): A missing case: Case 38, published by Hoëger et al., 2023

Response:  Case 38 remains the same. We have included Hoeger 2022 (thanks for calling our attention to it), which is now Case 39 (we have also corrected the statement on the number of cases in line 304).

- Figure 4: The repetition "activity activity" has not been corrected. We suggest separating the effects of pathogenic variants responsible for POIKTMP/Fibrosis from the effects responsible for cancers.

Response:  We have now corrected the "activity" error in the image. We appreciate the suggestion; however, we believe the image in its current form communicates the take-home message of the manuscript.